# Impact of Lung Ultrasound along with C-Reactive Protein Point-of-Care Testing on Clinical Decision-Making and Perceived Usefulness in Routine Healthcare for Patients with Lower Respiratory Tract Infections: Protocol for Analytical Observational Study

**DOI:** 10.3390/jcm13195770

**Published:** 2024-09-27

**Authors:** Anna Llinas, Eugeni Paredes, Joaquim Sol, Jose Maria Palacin, Monica Solanes, Javier Martinez, Carme Florensa, Laia Llort, Maria Teresa Castañ, Maria Isabel Gracia, Josep Miquel Paül, Marta Ortega Bravo, Carl Llor

**Affiliations:** 1Onze de Setembre Primary Care Centre, Institut Català de la Salut, 25005 Lleida, Spain; 2GRECOCAP Research Group, University Institute in Primary Care Research Jordi Gol, 08007 Barcelona, Spain; 3University Institute in Primary Care Research Jordi Gol, 08007 Barcelona, Spain; 4Centro de Investigación Biomédica en Red (CIBER) en Enfermedades Infecciosas, Instituto de Salud Carlos III, 28029 Madrid, Spain; 5Department of Medicine, Faculty of Medicine, University of Lleida, 25003 Lleida, Spain; 6Research Support Unit Lleida, University Institute in Primary Care Research Jordi Gol, 08007 Barcelona, Spain; 7Department of Experimental Medicine, University of Lleida-Biomedical Research Institute of Lleida (UdL-IRBLleida), 25003 Lleida, Spain; 8Balaguer Primary Care Centre, Institut Català de la Salut, 25600 Lleida, Spain; 9Borges Blanques Primary Care Centre, Institut Català de la Salut, 25400 Lleida, Spain; 10Rambla de Ferran Primary Care Centre, Institut Català de la Salut, 25007 Lleida, Spain; 11Cervera Primary Care Centre, Institut Català de la Salut, 25200 Lleida, Spain; 12Almacelles Primary Care Centre, Institut Català de la Salut, 25100 Lleida, Spain; 13Department of Public Health, General Practice, Faculty of Medicine, University of Southern Denmark, 5000 Odense, Denmark

**Keywords:** respiratory tract infections, point-of-care testing, primary healthcare, lung ultrasound

## Abstract

**Background**: Lower respiratory tract infections (LRTIs) are a significant cause of primary care consultations. Differentiating between viral and bacterial infections is critical for effective treatment and to minimize unnecessary antibiotic use. This study investigates the impact of combining lung ultrasound (LUS) with capillary blood C-reactive protein (CRP) rapid testing on clinical decision-making for patients with LRTIs. **Objectives**: The primary objective is to assess how the integration of CRP testing and LUS influences antibiotic prescription decisions. The study aims to quantify the percentage change in antibiotic prescriptions before and after performing LUS, following history taking, clinical examination, and CRP testing. **Methods**: This analytical observational study will be conducted in six primary care centers within the Health Region of Lleida, Catalonia, serving a mixed urban and rural population of approximately 105,000 residents. The study will recruit 196 patients aged 18 and over, presenting with LRTI symptoms and not treated with antibiotics in the preceding 14 days. Participants will undergo CRP testing followed by LUS during their consultation. Statistical analyses, including linear regression, bivariate analysis, Pearson or Spearman correlation, and logistic regression, will be used to evaluate the impact of LUS on clinical decisions and its predictive value in diagnosing bacterial infections. **Results and Conclusions**: This study will provide insights into the role of LUS combined with CRP testing in improving diagnostic accuracy and guiding antibiotic prescription decisions in primary care. The findings aim to enhance treatment protocols for LRTIs, reducing unnecessary antibiotic use and improving patient outcomes.

## 1. Introduction

Lower respiratory tract infections (LRTIs) are among the most common reasons for consultations in primary care centers [1]. According to the European Respiratory Society and the European Society for Clinical Microbiology and Infectious Diseases, LRTIs are defined as acute illnesses lasting less than 21 days, with a cough as the main symptom, accompanied by at least one other lower respiratory symptom, such as sputum production, dyspnoea, difficult breathing, or chest pain, without any other explanation for the clinical picture [2]. This category includes acute bronchitis, pneumonia, and acute exacerbations of chronic bronchitis and chronic obstructive pulmonary disease (COPD) [3].

Distinguishing between viral and bacterial causes of acute respiratory infections can be challenging. Despite the widespread use of antibiotics, it is estimated that 30–85% of these prescriptions are inadequate or unnecessary [4]. Spain currently has one of the highest rates of resistance to major respiratory pathogens [5]. Limiting antibiotic use to infections truly caused by bacteria is crucial due to increasing antimicrobial resistance [6]. The misuse of antibiotics does not lead to clinical improvement, exposes patients to higher risks of adverse effects, and increases the prevalence of resistant bacteria [7].

Adults who present to general practice with symptoms of acute LRTI often suffer from self-limiting viral infections. However, some patients have bacterial community-acquired pneumonia, a potential life-threatening infection, which requires immediate antibiotic treatment. Importantly, no single symptom or specific point-of-care test can be used to discriminate the various diagnoses, and diagnostic uncertainty often leads to the (over)use of antibiotics. At present, general practitioners lack tools to better identify those patients who will benefit from antibiotic treatment. The use of C-reactive protein (CRP) rapid testing has been associated with lower antibiotic prescribing rates. Clinicians using this rapid test can safely reduce antibiotic prescribing for LRTIs by 22% [8,9]. Ideally, rapid diagnostic techniques in primary care settings can reduce unnecessary antibiotic prescriptions in LRTIs by up to 70% [10,11]. Some clinical guidelines, including those from the European Guide to Infectious Diseases and Clinical Microbiology, recommend using CRP detection in capillary blood as a rapid diagnostic technique for managing LRTIs [2]. CRP is an acute-phase reactant produced by the liver in response to tissue damage or inflammation. Normal levels are below 5 mg/L but can increase to a maximum of 500 mg/L during acute inflammation [12]. In LRTIs, CRP levels below 20 mg/L suggest a self-limiting infection, presumably of viral origin, while levels above 100 mg/L indicate a severe bacterial infection [13]. Levels between 20 and 100 mg/L represent an area of uncertainty, where additional diagnostic tests would be beneficial [14]. For acute exacerbations of COPD, CRP values above 40 mg/mL suggest bacterial infections, with an uncertainty range of 20 to 40 mg/mL [15]. Point-of-care CRP testing has also been shown to improve antibiotic prescribing accuracy in patients with COPD exacerbations [16].

Lung ultrasound (LUS) has become a valuable complementary diagnostic tool since the SARS-CoV-2 pandemic in 2019 [17,18,19,20,21,22,23], and a valuable tool for identifying severe cases of COVID-19 [24]. Multiple studies indicate that it is more sensitive than the traditional chest X-ray for detecting and monitoring LRTIs [25,26], especially when there is pleural involvement [27]. LUS has demonstrated a higher sensitivity (95%) compared to chest X-rays (77%) for diagnosing adult community-acquired pneumonia (CAP), with specificities of 90% and 91%, respectively [28]. Subpleural consolidations smaller than 1.5 cm, multiple and bilateral, with pleural irregularities and bilateral B-lines, typically suggest a viral infection. In contrast, larger single consolidations, with bronchogram and pleural effusion, often indicate a bacterial infection [29,30,31,32].

LUS has several advantages; it does not expose patients to ionizing radiation, making it particularly useful for children, pregnant women, and patients requiring continuous monitoring [33,34]. Additionally, the devices are relatively inexpensive, facilitating their use in primary care centers. Some LUS devices are wireless, allowing for the possibility of performing ultrasounds at home [35].

Analyzing the perceived usefulness of utilizing the two diagnostic tests to enhance the diagnostic process and correlating LUS findings with CRP results would be valuable. While several studies have examined LUS patterns in LRTIs in children and intensive care units, there is a notable lack of research on employing this technique to differentiate between viral and bacterial etiologies in primary care settings.

Therefore, the main objective of this study is to determine whether adults who present with symptoms of an acute LRTI in general practice and who have performed LUS in addition to CRP are treated less frequently with antibiotics compared to the decision made following the completion of history taking, clinical examination, and CRP testing. Two secondary objectives will be analyzed: (a) to analyze the correlation between LUS findings and CRP results as a diagnostic tool for managing LRTIs; and (b) to determine the chronic conditions most frequently associated with LRTIs.

## 2. Materials and Methods

### 2.1. Study Design

This is a prospective observational study designed to assess the diagnostic utility of LUS combined with CRP testing in patients presenting with LRTIs in the primary care setting. The study will be conducted across six primary care centers within the Health Region of Lleida, Catalonia. 

### 2.2. Sample Size

Accepting an alpha risk of 0.95 for a precision of +/− 0.07 units in a two-sided test for an estimated rate of changes in antibiotic prescription of 0.5, based on preliminary results from a pilot study, 196 subjects randomly selected from the whole population are required. 

This sample size is also required to detect a minimum sensitivity for the detection of bacterial infection of 0.65 ± 0.15, with a confidence level of 95%. The formula used for the calculation can be found in [36].

### 2.3. Participants

Participants presenting with symptoms indicative of LRTIs will be selected through convenience sampling, coinciding with their visits to the primary care practices. 

To be eligible for the study, patients must fulfill all the inclusion criteria:Patients aged 18 years or older;Presenting with symptoms indicative of lower respiratory tract infections (LRTIs), defined as acute illnesses lasting less than 21 days, with a cough as the main symptom, accompanied by at least one other lower respiratory symptom, such as sputum production, dyspnea, difficult breathing, or chest pain, without any other explanation for the clinical picture [2];No antibiotic treatment in the past 14 days.

The presence of any of the following exclusion criteria leads to patient exclusion from the study:Hemodynamic instability;Recent thoracic surgery (in the last 60 days);Pre-existing conditions such as lung cancer, chronic pleural diseases or lung interstitial diseases.

### 2.4. Informed Consent

Eligible participants will be informed about the study and provided with an information leaflet and a consent form from the General Practitioner (GP). The GP will ensure that the patient understands the purpose of the project, potential benefits, risks, and procedures involved. The GP will underscore that participation is voluntary and that the patient can decline to participate or withdraw from the project at any time without consequences. The patient will be informed that consent to participate will give the primary investigator, sponsor, and controlling authorities access to obtain information in the patient’s medical records, including the electronic records, to obtain information about the patient’s health conditions. Due to the nature of the project, the patient will only be given a few minutes to consider participation.

The GP will check patients against the eligibility criteria stated above and invite patients to participate if they fulfill all the inclusion criteria and none of the exclusion criteria. Patients who agree to participate will be asked to provide written consent, which will be obtained by the GP. Informed consent will be obtained prior to the collection of participant data. Participants will be informed about the storage and use of their data.

### 2.5. Interventions

#### 2.5.1. Intervention Description

The recruitment process for participants will commence simultaneously across six primary care centers located within the Health Region of Lleida, Catalonia. To ensure a diverse and representative sample, participants will be selected through convenience sampling. This method will involve approaching patients during their routine visits to the primary care practices, including both scheduled appointments and emergency consultations, until the target sample size is reached.

The attending healthcare professional who first visits the patient will diagnose LRTI according to the definition outlined in the introduction. A standardized diagnostic approach will be used across all six participating primary care centers, ensuring that every healthcare professional follows the same guidelines for diagnosing LRTIs. If the patient meets the pre-established inclusion criteria, the attending healthcare professional will refer the patient to the designated recruiting investigator. The recruiting investigator will then explain the study details to the patient, including its purpose, procedures, and potential risks and benefits. If the patient agrees to participate, they will be provided with an Informed Consent form and a Patient Information Sheet. These documents will outline the study’s objectives, the nature of the interventions, and the participant’s rights.

The intervention process will begin with a thorough collection of the patient’s clinical history. The GP will review and document key aspects of the patient’s medical background, including any current symptoms and previous medical conditions.

Following the medical history, the GP will conduct a detailed physical examination. This examination will include the assessment of vital signs such as temperature, heart rate, respiratory rate, and blood pressure. The GP will also evaluate specific symptoms reported by the patient, such as signs of respiratory distress, and will assess the respiratory function. The physical examination aims to provide a baseline assessment and to identify any immediate clinical concerns.

Subsequently, a primary care nurse will perform the CRP test. This involves a small puncture in the patient’s fingertip to obtain a drop of blood. The blood sample will be analyzed for CRP levels, a marker of inflammation, which typically takes less than 10 min to process. The CRP test results will provide valuable information on the level of systemic inflammation. 

Afterward, the GP will carry out a LUS examination to evaluate lung pathology. The LUS will be performed using a portable ultrasound device, and the examination is expected to take less than 30 min. The GP will use this imaging technique to assess for signs of consolidations, pleural effusion, and other pulmonary abnormalities. Ultrasound images obtained during the study will be carefully reviewed and documented in the electronic health record, with each image accurately linked to the patient’s medical record.

During the intervention process, the GP will assess the need for antibiotic treatment at two key stages. Initially, prior to performing the LUS, the GP will make a preliminary evaluation of whether antibiotics might be necessary based on the patient’s clinical history, physical examination, and the initial CRP test results.

After completing the LUS, the GP will review the additional diagnostic information obtained from the ultrasound imaging. The LUS findings, combined with the CRP test results and the clinical assessment conducted earlier, will provide a comprehensive view of the patient’s respiratory status. This post-LUS evaluation allows the GP to reconsider the initial antibiotic prescription decision based on the full set of diagnostic data.

The GP will integrate the findings from both the CRP test and the LUS to make an informed decision about whether antibiotic therapy is warranted. The decision-making process aims to ensure that antibiotic prescriptions are based on a thorough evaluation of all relevant clinical and diagnostic information, thereby optimizing treatment decisions and minimizing unnecessary antibiotic use.

Patients will be followed up after the medical visit in accordance with standard clinical practice to monitor their progress. Additionally, a telephone evaluation will be conducted 30 days post-visit to assess for any complications or worsening of LRTI.

The overall process of the study is illustrated in Figure 1, which provides a detailed flow chart of the study design and intervention steps.

#### 2.5.2. CRP Point-of-Care Testing Training Program

A one-hour training workshop will take place for all participating physicians before the inception of the study with an explanation of the clinical guidelines. Participating professionals will be informed about the updated recommendations on how to interpret the CRP values and the evidence-based management. 

#### 2.5.3. CRP Point-of-Care Testing Interpretation

CRP levels will be interpreted as follows:A total <20 mg/L indicates a self-limiting infection and >100 mg/L suggests a bacterial infection, with 20–100 mg/L indicating uncertainty.For COPD exacerbations, <20 mg/L hints at a self-limiting infection and >40 mg/L points to a bacterial infection.

#### 2.5.4. LUS Training Program

To minimize inter-observer variability in performing the LUS, all participating physicians will undergo a training session prior to the study’s commencement. All GPs participating in this study have already taken validated LUS training programs before.

This training will focus on standardizing the LUS procedure to ensure consistency and reliability across all practitioners. The LUS training program will consist of a two-hour course focused on both theoretical and practical components. The training will begin with an introduction to LUS, covering the fundamental principles, clinical applications, and importance of LUS in primary care settings.

Following the introduction, the course will include a revision of normal and pathological LUS images, enabling participants to differentiate between healthy lung tissue and various pathological conditions such as pneumonia, pleural effusions, and pneumothorax. 

To ensure consistency and accuracy in LUS practice, the course will also include a session dedicated to the standardization of the LUS technique. During this session, participants will focus on standard procedures such as patient positioning and systematic scanning order. GPs will practice the technique to ensure that LUS is performed consistently, allowing for reliable and reproducible results across different clinical settings.

The hemithorax will be divided into three surfaces: anterior, lateral, and posterior. The anterior and lateral surfaces will be further subdivided into upper and lower quadrants on each side, while the posterior surface will be divided into upper, middle, and lower quadrants. Each of these quadrants will constitute a distinct scanning zone. On the patient’s left side, the zones will be labeled from 1 to 7L, and on the right side, they will be labeled from 1 to 7R. GPs will be thoroughly trained in this standardized 14-zone scanning technique as part of the LUS training program.

By the end of the training, GPs will be expected to perform LUS examinations in a standardized manner, interpret both normal and pathological findings accurately, and integrate these findings into their clinical decision-making process.

#### 2.5.5. LUS Images Interpretation

The LUS pathological findings are predefined and GPs will be trained during the LUS program to identify and interpret these images. The definitions of the LUS findings are derived from the European Federation of Societies for Ultrasound in Medicine and Biology (EFSUMB) coursebook [37].

B-lines: laser-like vertical echogenic artifacts arising from the pleural line, spreading without fading to the edge of the screen and moving synchronously with lung sliding.Interstitial syndrome: multiple (≥3) B-lines in at least 2 zones on each side present.Consolidation: loss of aeration, which allows the visualization of the lung parenchyma with sonomorphologic characteristics that resemble a solid organ or tissue. Pathognomonics for a pneumonic consolidation is the presence of air bronchograms and serrated or blurred margins.Subpleural consolidation: small subpleural consolidation between 2 and 20 mm in size that moves together with lung sliding.Pleural effusion: anechoic or hypoechoic space between the visceral and parietal pleura.Focal visceral pleural pathology: hypoechogenic thickening of the pleura with a rough appearance and interruption of the normally smooth pleura.Pneumothorax: area without lung sliding, lung pulse or B-lines, with the presence of a lung point in an adjacent area.Air bronchogram: multiple hyperechoic densities surrounded by hypoechoic consolidated lung parenchyma.Other LUS images: other findings by LUS will be described according to the GPs ability.

### 2.6. Data Collection

Data will be collected at the point of initial patient contact through direct medical assessments, which include LUS and CRP testing. Additional data will be gathered from electronic health records, capturing demographic information and clinical signs and symptoms at the time of presentation.

Clinical interview: age, sex, comorbidities (hypertension, diabetes mellitus, dyslipidemia, chronic kidney disease, chronic liver disease, asthma, chronic obstructive pulmonary disease, pulmonary fibrosis, other chronic pulmonary diseases), current immunosuppressive drugs.

Current respiratory symptoms: days since symptoms onset, fever, cough, sputum production, dyspnea, chest pain, sputum coloration, rhinorrhea, respiratory difficulty.

Physical examination: temperature, heart rate, respiratory rate, blood pressure, oxygen saturation, tachypnea, tachycardia, asymmetric chest expansion, use of accessory muscles in breathing, rhonchi, crackles, wheezing, egophony, hypophonia, dullness on percussion.

Measurement of the C-Reactive Protein (CRP) value.

Ultrasound Findings: pleural effusion, pleural sliding, presence of consolidation, size of the main consolidation, number of consolidations, location of consolidations, presence of bronchogram, B-line pattern.

Final Diagnosis: viral or self-limiting infection, bacterial infection.

Antibiotic Therapy Prescription: antibiotic prescription decision prior to pulmonary ultrasound, antibiotic prescription decision after pulmonary ultrasound, final antibiotic prescription decision. 

Follow-up: worsening of symptoms, persistence of symptoms, resolution of symptoms, hospitalization, antibiotic prescription posterior to the medical visit.

Patients will be informed about the type of data that will be collected and about their rights when being informed about the study and will sign a consent form consenting the use of their data. No data will be collected before the patient signs the consent form. As soon as the patient has signed the consent form, they will be given a study code and will be identified by the study code in all the study material. The relationship between their identity and the study code will be kept in the healthcare center and will only be available to those that need it to fulfill their duties. Data collected in this study are considered codified data.

### 2.7. Data Storage

The participant lists will be stored electronically and protected using identification codes with data dissociation in the Research Electronic Data Capture (REDCap) version 14.3.14. IDIAPJGol (University Institute in Primary Care Research Jordi Gol) has license to use REDCap and the data are collected and held in IDIAPJGol servers, which comply with the European and national requirements. The encoded database will be accessible and will only be used by the research team and collaborating teams working on this study. The transfer of data from the collected database to REDCap for data analysis will be conducted confidentially and anonymously. The data will be stored for 10 years in the REDCap and will be subsequently destroyed. Given that the Primary Care Clinical Station (ECAP) program already maintains a registry for storing the results of capillary blood PCR tests and LUS results, patient data will be retained in their medical records indefinitely, following standard healthcare practice.

The results from this study are intended for publication in peer-reviewed international journals and will be distributed across the network of involved healthcare facilities.

### 2.8. Outcomes

The primary outcome will be the initial decision to prescribe antibiotics before performing LUS, the influence of ultrasound findings on the likelihood of subsequent antibiotic prescription, and the changes in antibiotic prescription rates pre- and post-ultrasound procedure.

Secondary outcomes will include the correlation between LUS findings and CRP levels as a diagnostic tool for managing LRTIs, measuring the correlation between numerical values of CRP and LUS results indicative of viral or bacterial etiologies.

The identification and documentation of chronic conditions most frequently associated with LRTIs in the study population will also be evaluated.

### 2.9. Statistical Analyses

The study sample will be described using standard statistical methods based on the type and distribution of the variables. The effect of using LUS on the clinical decision-making of physicians will be examined. A linear regression will be calculated to predict the number of clinical decision changes following LUS performance after the use of CRP rapid testing, considering the work experience at the physician level. In this analysis and in the calculation of 95% CIs, clustering at the physician level will be accounted for.

Subsequently, bivariate analysis will be conducted to assess the correlation between CRP levels and LUS findings. For continuous CRP levels versus categorical LUS findings, Student’s t-tests or Mann–Whitney U tests will be employed. For categorical CRP levels versus categorical LUS findings, chi-square tests or Fisher’s exact tests will be used. For continuous CRP levels versus continuous LUS findings, Pearson or Spearman correlation coefficients will be calculated as appropriate. All analyzed variables will be included in these analyses. The independent predictive ability of LUS findings for identifying bacterial LRTIs will be evaluated while controlling for potential confounders using logistic or linear regressions, considering the continuous and categorical CPR levels as the response variables, respectively. The results will be reported as regression estimates with their 95% confidence intervals. Model performance will be assessed using goodness of fit tests. The area under the receiver operating characteristic (ROC) curves, along with the sensitivity, specificity, positive predictive value (PPV), and negative predictive value (NPV) will be calculated to assess the diagnostic accuracy of LUS findings in cases confirmed with CRP.

The influence of age and patients’ comorbidities will be considered either by stratifying the analyses by these variables or by including them in the models, together with their interaction with the LUS results.

Analyses will be conducted using R version 4.1.2, with statistical significance set at *p* < 0.05.

### 2.10. Ethical Approval

Ethical approval for the study has been provided by the Research Ethics Committee of the University Institute in Primary Care Research Jordi Gol (23/222-P). The study will be carried out following the recommendations of the Declaration of Helsinki. The data and variables collected from the participants will be treated anonymously and confidentiality will be guaranteed. The evaluation will be conducted in compliance with Regulation 2016/679 of the European Parliament and of the Council, dated 27 April 2016, pertaining to the protection of individuals with respect to the processing of personal data and the unimpeded flow of said data. The evaluation will also adhere to Organic Law 3/2018, dated 5 December 2018, on the Protection of Personal Data and Guarantee of Digital Rights.

## 3. Discussion

### 3.1. Potential Impact and Significance of the Study

This study highlights the utility of combining LUS with CRP rapid testing in managing LRTIs. The primary aim is to evaluate the decision to prescribe antibiotics before performing LUS and how it influences subsequent antibiotic prescriptions, which has not been extensively studied before. Previous studies utilized LUS-CRP correlation mainly to assess infection severity, particularly in COVID-19 patients [38,39]. Similarly, another study demonstrated the diagnostic accuracy and clinical correlations of LUS in COVID-19 pneumonia but focused on acute care settings [40]. This study, in contrast, targets a primary care environment and emphasizes the diagnostic interplay between LUS and CRP to differentiate bacterial from viral infections, aiding in clinical decision-making and probably reducing unnecessary antibiotic prescriptions.

The findings support that LUS, combined with CRP testing, offers a robust diagnostic framework, enhancing the accuracy of diagnosing LRTIs. This integrated approach is particularly valuable in primary care, where rapid and non-invasive diagnostic tools are essential for effective patient management. This study’s comprehensive approach, combining two diagnostic modalities, addresses a significant gap in the literature by providing evidence of the practical utility of LUS and CRP in routine clinical settings.

In conclusion, implementing these diagnostic tools in primary care can significantly reduce inappropriate antibiotic prescriptions, helping physicians make more robust and informed decisions regarding the prescription of antibiotic therapy, which aligns with the global effort to combat antimicrobial resistance.

### 3.2. Limitations

One of the main limitations that we forecast in this study is the inter-observer variability in the performance of LUS. For this reason, a previous training program for all the GPs participating in the study will be carried out in order to minimize this limitation, although there may still be significant variability between different physicians’ interpretations of LUS findings.

Regarding CRP rapid testing, we recognize that elevated CRP levels can signal systemic inflammation in conditions beyond infections, such as inflammatory disorders. Moreover, there is limited research on how CRP levels fluctuate daily in LRTIs. CRP levels above 40 mg/L might represent the natural progression of an infection, particularly depending on when the patient consults at the primary care center.

The absence of a control group without performing LUS could be seen as a limitation. However, this study follows a self-controlled design, where medical decisions are first evaluated without the use of LUS (serving as the ‘control’), followed by a reassessment after LUS is performed (serving as the ‘intervention’). While this approach strengthens the internal comparison, it is still not possible to entirely rule out the influence of other factors on prescribing behavior.

The use of convenience sampling introduces the risk of selection bias, as the sample may not be fully representative. Additionally, since physicians are not blinded to CRP results during initial prescribing decisions, their judgment may be influenced, potentially impacting the objectivity of the outcomes.

The study is conducted in a specific region of Catalonia, which may limit the generalizability of the results to other healthcare systems or populations. Variations in clinical practices, patient demographics, and healthcare resources may affect the applicability of the findings elsewhere.

While there is a risk that physicians may become overly dependent on LUS and CRP results, it is important to note that a thorough clinical history and physical examination will be performed prior to utilizing these diagnostic tools. Although the use of LUS and CRP is designed to complement clinical judgment, rather than replace the evaluation of key symptoms and signs, this potential for over-reliance remains a limitation of the study.

The potential value of conducting a cost-effectiveness analysis is recognized, particularly given the resource implications of implementing LUS in primary care. However, this aspect falls outside the scope of the current study and would be an important focus for future research.

The awareness among physicians that they are being studied may influence their prescribing behavior, potentially leading to more cautious or altered decision-making. This could result in an overestimation of the impact of the diagnostic tools, as their actions may not fully reflect routine clinical practice.

## Figures and Tables

**Figure 1 jcm-13-05770-f001:**
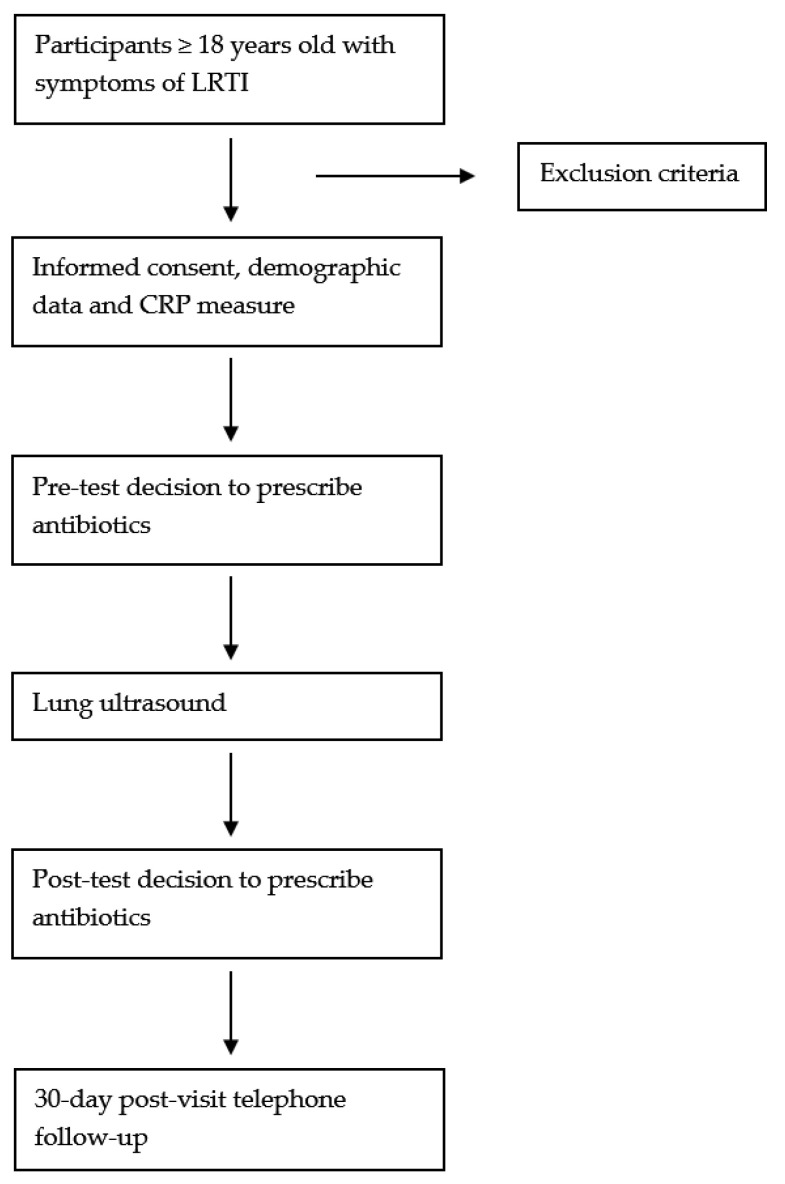
Flow chart of the study.

## Data Availability

For information about this study, please contact the corresponding author. At present, data access is restricted as participant enrolment and consent collection have not yet begun. Additionally, our research is focused on methodological aspects, and we do not have any clinical data related to the patients at this time.

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
