# Peer review of "Impact of Lung Ultrasound along with C-Reactive Protein Point-of-Care Testing on Clinical Decision-Making and Perceived Usefulness in Routine Healthcare for Patients with Lower Respiratory Tract Infections: Protocol for Analytical Observational Study"

_jcm, 2024, doi:10.3390/jcm13195770_

Round 1
Reviewer 1 Report
Comments and Suggestions for Authors
Dear authors,
I have now completed the review of the manuscript titled "Impact of lung ultrasound along with C-reactive protein point-of-care testing on clinical decision-making and perceived usefulness in routine healthcare for patients with lower respiratory tract infections: protocol for an analytical observational study."
In the present study, the authors used well-defined objectives, therefore the study has clear primary and secondary objectives focused on evaluating the impact of lung ultrasound (LUS) and C-reactive protein (CRP) testing on antibiotic prescribing decisions for lower respiratory tract infections (LRTIs). Combining LUS with CRP testing may provide more comprehensive diagnostic information than either tool alone.
Also, the protocol outlines a detailed process for patient recruitment, data collection, and analysis, including standardized training for physicians performing LUS. If successful, this study could provide valuable evidence for improving antibiotic stewardship in primary care settings.
The manuscript is interesting and, in general, fairly well-written.
I have some suggestions to further improve the quality of the manuscript.
I would like to suggest that the authors address these limitations in the article, either by discussing them in the limitations section or, where feasible, by making the appropriate revisions:
1. Sample size justification: While a sample size calculation is provided, it's unclear if this is adequately powered for all the study's objectives, particularly the secondary outcomes.
2. Lack of control group: The study design does not include a control group not receiving LUS, which limits the ability to definitively attribute changes in prescribing behavior to the use of LUS.
3. Potential for bias: The study relies on convenience sampling, which may introduce selection bias. Additionally, physicians are not blinded to CRP results when making initial prescribing decisions, potentially influencing their judgment.
4. Inter-observer variability: While the protocol includes training to standardize LUS technique, there may still be significant variability between different physicians' interpretations of LUS findings.
5. Limited generalizability: The study is conducted in a specific region of Catalonia, and results may not be generalizable to other healthcare systems or populations.
6. Lack of long-term follow-up: The study does not include plans for following up on patient outcomes, which could provide valuable information on the clinical impact of prescribing decisions.
7. Potential for overreliance on diagnostic tools: There's a risk that physicians may become overly dependent on LUS and CRP results, potentially overlooking other important clinical factors.
8. Limited exploration of patient factors: The study could benefit from more in-depth analysis of how patient characteristics (e.g., age, comorbidities) influence the utility of LUS and CRP in guiding prescribing decisions.
9. Absence of cost-effectiveness analysis: Given the resource implications of implementing LUS in primary care, an economic evaluation component could strengthen the study's impact.
10. Potential for Hawthorne effect: Physicians' awareness of being studied may influence their prescribing behavior, potentially overestimating the impact of the diagnostic tools.
Thank you for your valuable contributions to our field of research. I look forward to receiving the revised manuscript.
Author Response
Dear Reviewer,
Thank you for your detailed and thoughtful review of our manuscript. We greatly appreciate your feedback, which has helped us improve the quality of our study. Please find our responses to your comments below.
Comment 1: Sample size justification: While a sample size calculation is provided, it's unclear if this is adequately powered for all the study's objectives, particularly the secondary outcomes.
Response 1: The sample size calculation has been clarified (page 3), and an additional bibliographic reference has been included to support this (reference 37).
Comment 2: Lack of control group: The study design does not include a control group not receiving LUS, which limits the ability to definitively attribute changes in prescribing behavior to the use of LUS.
Response 2: The study design is self-controlled, where decisions are first made without LUS ("control") and then after LUS ("intervention"). This design aspect has now been emphasized in the limitations section to address this point (page 10, second paragraph).
Comment 3: Potential for bias: The study relies on convenience sampling, which may introduce selection bias. Additionally, physicians are not blinded to CRP results when making initial prescribing decisions, potentially influencing their judgment.
Response 3: The risk of selection bias from convenience sampling, as well as the potential influence of CRP results on prescribing behaviour, have been acknowledged and highlighted as limitations (page 10, third paragraph).
Comment 4: Inter-observer variability: While the protocol includes training to standardize LUS technique, there may still be significant variability between different physicians' interpretations of LUS findings.
Response 4: While there may be inter-observer variability in LUS interpretation, standardized training will be provided to minimize this issue. This has been clarified further in the limitation section, though it remains a limitation (page 9, first paragraph in limitations).
Comment 5: Limited generalizability: The study is conducted in a specific region of Catalonia, and results may not be generalizable to other healthcare systems or populations.
Response 5: This point has been added to the limitations section (page 10, fourth paragraph), noting that the study is conducted in a specific region, and the results may not be generalizable to other healthcare settings.
Comment 6: Lack of long-term follow-up: The study does not include plans for following up on patient outcomes, which could provide valuable information on the clinical impact of prescribing decisions.
Response 6: To address this concern, it has been specified that patients will be followed up after their medical visit in accordance with standard clinical practice to monitor progress. Additionally, a telephone evaluation will be conducted 30 days post-visit to assess for any complications or worsening of LRTI. This follow-up plan has been added to the intervention description section (page 5, third paragraph) and the study flow chart (Figure 1) has been updated to include the 30-day telephone follow-up. The data collection section has also been revised to reflect this addition (page 8, first paragraph).
Comment 7: Potential for overreliance on diagnostic tools: There's a risk that physicians may become overly dependent on LUS and CRP results, potentially overlooking other important clinical factors.
Response 7: This concern has been acknowledged and included as a limitation in the study (page 10, fifth paragraph). However, it has been emphasized that the use of LUS and CRP is always preceded by a thorough clinical assessment, including detailed history-taking and physical examination. These diagnostic tools are intended to complement, not replace, clinical judgment.
Comment 8: Limited exploration of patient factors: The study could benefit from more in-depth analysis of how patient characteristics (e.g., age, comorbidities) influence the utility of LUS and CRP in guiding prescribing decisions.
Response 8: A line has been added to the statistical analysis section (page 9, second paragraph) to include that patient characteristics, such as age and comorbidities, will be considered either by stratifying the analyses by these variables or by including them in the models.
Comment 9: Absence of cost-effectiveness analysis: Given the resource implications of implementing LUS in primary care, an economic evaluation component could strengthen the study's impact.
Response 9: The value of a cost-effectiveness analysis has been acknowledged in the limitations section (page 10, sixth paragraph). This is recognized as a potential area for future research.
Comment 10: Potential for Hawthorne effect: Physicians' awareness of being studied may influence their prescribing behavior, potentially overestimating the impact of the diagnostic tools.
Response 10: The possibility of a Hawthorne effect, where physicians’ awareness of being studied may influence prescribing behaviour, has been included as a limitation (page 10, seventh paragraph).
Once again, we greatly appreciate your insightful comments, and we hope that the revisions we have made address your concerns. We look forward to your further consideration of our manuscript.
Yours sincerely,
Anna Llinas
Reviewer 2 Report
Comments and Suggestions for Authors
Comments to the Author
The author proposed the protocol to analyze diagnostic accuracy for LITI by the combination with CRP and LUS. This paper is well written but has the following problems.
Major comments
1. Please provide more details how you diagnosed with LRTI as follows.
・Were LRTI diagnoses made by each attending physician? If so, was a central diagnosis perform to correct for discrepancies in diagnostic content?
・In this study, is not excluding any underlying chronic respiratory disease such as fibrotic interstitial lung disease, chronic obstructive lung disease, chronic pleural disease (e.g. asbestosis) needed? Is not those underlying disease expected to influence study result.
・Is exclusion criteria about treated drugs set? I worry the possibility that immunosuppressive drugs such as corticosteroid reduce CRP levels.
2. Among blood test, is not differentiating ability between viral and bacterial infection, by increase of white blood cells, neutrophil and presence of poisoning granules than CRP? Are not those variable needed?
Author Response
Dear Reviewer,
We would like to thank you for your thorough review of our manuscript and the time you’ve invested in providing such valuable feedback. Please find our responses to your comments below.
Comment 1: Were LRTI diagnoses made by each attending physician? If so, was a central diagnosis perform to correct for discrepancies in diagnostic content?
Response 1: It has been clarified in the description process section (page 4, second paragraph) that LRTI diagnoses are made by the attending physician at the primary care center according to a standardized diagnostic approach. This unified diagnostic approach has been implemented across all collaborating primary care centers, based on the LRTI definition established in the literature.
Comment 2: In this study, is not excluding any underlying chronic respiratory disease such as fibrotic interstitial lung disease, chronic obstructive lung disease, chronic pleural disease (e.g. asbestosis) needed? Is not those underlying disease expected to influence study result.
Response 2: The exclusion criteria have been modified to exclude lung interstitial and chronic pleural diseases, which can potentially affect the LUS images observed (page 3, third exclusion criteria point). However, COPD exacerbation will be included, as this population is relevant in the context of LRTIs and we would like to evaluate the utility of LUS and CRP in this clinical setting.
Comment 3: Is exclusion criteria about treated drugs set? I worry the possibility that immunosuppressive drugs such as corticosteroid reduce CRP levels.
Response 3: It has been noted that this will be included as a variable in the data collection section rather than as an exclusion criterion (page 7, second paragraph). Patients receiving immunosuppressive drugs, such as corticosteroids, could represent a subgroup where LUS may be particularly useful, as they might have low CRP levels while actually having a bacterial infection.
Comment 4: Among blood test, is not differentiating ability between viral and bacterial infection, by increase of white blood cells, neutrophil and presence of poisoning granules than CRP? Are not those variable needed?
Response 4: We would like to emphasize that this study is conducted in a primary care setting where urgent blood analyses are not feasible. Therefore, the use of CRP as a rapid diagnostic test has been highlighted in this scenario for its proven utility in numerous studies.
Once again, we deeply appreciate your helpful comments and have revised the manuscript accordingly. We hope the changes address your concerns and that you will consider our article for publication.
Thank you for your time and consideration.
Yours sincerely,
Anna Llinas
Round 2
Reviewer 1 Report
Comments and Suggestions for Authors
All comments were addressed.
Reviewer 2 Report
Comments and Suggestions for Authors
Author responsed to queries correctly.